# Advancing Kidney Transplantation: A Machine Learning Approach to Enhance Donor–Recipient Matching

**DOI:** 10.3390/diagnostics14192119

**Published:** 2024-09-25

**Authors:** Nahed Alowidi, Razan Ali, Munera Sadaqah, Fatmah M. A. Naemi

**Affiliations:** 1Department of Computer Science, Faculty of Computing and Information Technology, King Abdulaziz University, Jeddah 21589, Saudi Arabia; 2Histocompatibility and Immunogenetics Laboratory, King Fahad General Hospital, Jeddah 21589, Saudi Arabia; fnaemi@moh.gov.sa

**Keywords:** kidney transplantation, donor–recipient matching, machine learning, intelligent allocation

## Abstract

(1) Background: Globally, the kidney donor shortage has made the allocation process critical for patients awaiting a kidney transplant. Adopting Machine Learning (ML) models for donor–recipient matching can potentially improve kidney allocation processes when compared with traditional points-based systems. (2) Methods: This study developed an ML-based approach for donor–recipient matching. A comprehensive evaluation was conducted using ten widely used classifiers (logistic regression, decision tree, random forest, support vector machine, gradient boosting, boost, CatBoost, LightGBM, naive Bayes, and neural networks) across three experimental scenarios to ensure a robust approach. The first scenario used the original dataset, the second used a merged version of the dataset, and the last scenario used a hierarchical architecture model. Additionally, a custom ranking algorithm was designed to identify the most suitable recipients. Finally, the ML-based donor–recipient matching model was integrated into a web-based platform called Nephron. (3) Results: The gradient boost model was the top performer, achieving a remarkable and consistent accuracy rate of 98% across the three experimental scenarios. Furthermore, the custom ranking algorithm outperformed the conventional cosine and Jaccard similarity methods in identifying the most suitable recipients. Importantly, the platform not only facilitated efficient patient selection and prioritisation for kidney allocation but can be flexibly adapted for other solid organ allocation systems built on similar criteria. (4) Conclusions: This study proposes an ML-based approach to optimize donor-recipient matching within the kidney allocation process. Successful implementation of this methodology demonstrates significant potential to enhance both efficiency and fairness in kidney transplantation.

## 1. Introduction

Kidney transplantation is the only curative therapy for patients with end-stage renal failure [1,2]. It has a beneficial impact on returning patients’ lives to normal and reducing mortality rates, particularly for younger patients [3]. When compared with other organs, kidney transplantation is the most common type of solid organ transplantation, accounting for 65% of all transplant cases worldwide. This is attributed to a steady increase in patients with end-stage renal failure, which is expected to increase to more than 5 million global cases by 2030 [4]. From recent statistics, the number of patients awaiting transplantation in the United States reached 110,000, and in Europe, this number was 14,000, with an increase of one case every 10 min. In the Kingdom of Saudi Arabia, dialysis patients exceeded 20,000 according to the Saudi Centre of Organ Transplantation (2021), with a prevalence rate of 621 persons/million individuals [5]. An increase in patients with end-stage renal failure has been estimated at an annual 5% [5]. Globally, the number of patients awaiting kidney transplantation has now exceeded the number of transplantation donors; therefore, organ shortages are the main organ transplantation obstacles. Kidneys for transplantation can be retrieved from either deceased (brain death or circulatory arrest) or living donors. Thus, due to resource scarcity, the implementation of an efficient transplantation system optimising organ utility is mandatory.

Internationally, organ donation and transplantation programmes are well defined and governed by local organ transplantation laws and legislation. The main principle of such laws, which were established in the United States in 1984, is the mandatory assurance of equity and justice between all patients irrespective of their ethnicity or geographical location. Kidney allocation is a fundamental part of any kidney transplant programme from a deceased donor. The allocation should combine the main ethical principles of transplantation, which are transparency, justice, utility, and equity of access for all patients [6]. Kidney allocation includes allocating kidneys from a deceased donor/altruistic living donor to the best-matched recipients previously registered on national waiting lists. A best match donor–recipient pair selection occurs according to pre-defined allocation rules and policies. The kidney allocation system in the United States (since 2014), the new kidney offering scheme in the UK, and the Eurotransplant kidney allocation programme are all well-structured kidney allocation systems from deceased donors [7,8,9]. These systems are based on points or scoring schemes using defined criteria and rules, with minor feature variations in each allocation scheme. A points-based system uses donor and recipient characteristics to set algorithms for allocation, and recipients achieving higher scores will be eligible to receive the offered kidney [10]. Generally, features such as donor–recipient age matching, waiting time, blood group compatibility, patient location, transplant urgency, human leukocyte antigen (HLA) matching, and HLA sensitisation are specified in most points-based kidney allocation systems [10]. The variables used in any allocation system and their prioritisation are based on recommendations from an annual review of transplant activity that is performed locally in most countries with well-defined transplant programmes [11]. Therefore, these systems are regularly subjected to continuous modifications and exceptions to maintain equity and utility. A notable modification in many kidney allocation systems is the inclusion of a longevity matching principle, which relies on age differences between a donor and recipient. This measure facilitates optimal graft utilisation, as it circumvents the demise of recipients with a functioning graft owing to their diminished estimated life expectancy and the prevention of re-transplantation for a younger patient because of inferior graft survival [12,13,14]. In this term, many allocation algorithms apply age-matching criteria between donor–recipient pairs such that a young recipient receives a young graft, and an older recipient receives a similar graft. For example, the kidney allocation scheme in British Columbia (Canada) allocates kidneys from donors less than 35 years old to recipients under 55 years old, providing an acceptable 20-year age difference between potential pairs [15]. In most developing countries, national kidney allocation programmes are not well defined, with many transplant centres adopting manual, unsophisticated systems for donor–recipient pair matching when a deceased donor becomes available. Such allocation systems have many drawbacks, such as prolonged times for recipient selection, loss of optimal utility, biases, and a lack of fairness and equity, all of which may negatively influence transplant outcomes and erode public confidence in transplant programmes [16,17]. Therefore, an allocation system that efficiently selects recipients in a timely and fair manner is warranted.

Blood group matching, HLAs, and HLA antibodies are the main biological factors that are first considered in most allocation algorithms, with slight differences between systems according to a country’s experience. HLA molecules are one of the most polymorphic genetic systems, with more than 30,000 alleles reported since 2023 [18]. Two HLA classes are primarily implicated in graft acceptance or rejection; HLA class I antigens constitute HLA-A, B, and C, and HLA class II antigens constitute DR (alpha and B chains) and DQ (alpha and B chains), which are inherited as a haplotype (A, B, C, DR, and DQ) from each parent [18]. The degree of HLA matching between a donor and recipient impacts graft and recipient survival. In general, the better the HLA match, the better the allograft survival, and the lower the requirement for immunosuppression [19]. In many countries, compatibility at HLA-A, B, and DR antigen levels is primarily considered when assessing the matching degree between a deceased donor and recipient; this impacts graft survival, with differences in priority ranking observed between different allocation systems [20,21]. Zero mismatch between a donor and recipient at A, B, and DR HLA levels is superior in terms of graft survival when compared with 3/4/6 HLA mismatches [22]. Additionally, donor HLA-C and HLA-DQ antigen acceptance is based on assessing the absence of HLA antibodies in recipients against corresponding donor HLA antigens. This assessment step is performed manually in most allocation systems, thereby prolonging organ allocation time and increasing the possibility of human error. Therefore, donor graft acceptance based on HLA antibodies must be performed for all HLA antigens to avoid serious hyperacute graft rejection complications. All steps in a matching process must be performed efficiently and in timely manner to preserve graft function after transplantation. For certain organs such as the kidneys, the transplant must proceed in a 12–24 h window, including organ allocation time, to avoid a delayed graft function [23]. Therefore, the development of an intelligent system for donor–recipient HLA matching could help optimise the allocation process by reducing allocation times, eliminating human error, and maintaining fairness.

ML is a part of artificial intelligence where a computer system uses mathematical algorithms, learned from previous known data, to generate similar findings and relationships when unknown data are entered [24]. ML has been used in various medical fields to analyse large and complex datasets to identify different relationships. This advantage motivates scientists to develop models that help facilitate disease diagnostics and predict outcomes with impressive accuracy [25]. Organ transplantation is one such field where ML may help analyse the huge number of variables recorded for donors and recipients. ML models have been developed for different transplanted solid organs, including the kidneys, liver, heart, and lungs [26,27,28,29]. Several models have been developed in the kidney transplantation field to assess both graft and patient survival rates, as well as for post-transplant management [30]. The most common classifiers are artificial neural networks (ANNs), convolutional neural networks, decision trees, gradient descent boosting, random forest, and logistic regression models with LASSO regularisation. Bae et al. (2019) developed an online model that used a random survival forest to estimate patient survival rates after kidney transplantation using datasets consisting of post-transplant survival scores, co-morbidities, time on dialysis, and previous transplant history [31]. Divard et al. (2022) showed comparable results between a physician’s prediction of a kidney graft failure and the predictive score of a designed prediction model [32]. Another model, developed by Luo et al. (2020), predicted post-transplant complications with 0.97 specificity using a random forest algorithm and a dataset consisting of 519 recipients with severe pneumonia [33]. Predicting delayed graft function after kidney transplantation was also assessed by developing a prognostic model using data from 55,000 recipients and multiple ML algorithms [34]. In a later study, an ANN model showed superior performance when compared with other models. Using datasets collected from more than 14,000 previous donor biopsies, Yoo et al. (2024) developed a model that assessed donor kidney quality to discriminate between donor graft injury and injury due to recipient factors [35]. Additionally, different models have been developed to optimise immunosuppression medication for post-kidney transplants [36,37]. However, no ML models have been developed for donor–recipient matching in kidney transplantation, which may improve kidney allocation processes.

In this study, an intelligence-based ML model was established that efficiently assessed donor–recipient pair matching in terms of ABO matching, HLA matching, HLA antibody compatibility, and age suitability. The dataset consisted of 10 HLA molecules (A, B, C, DR, and DQ) that were typed at two-digit molecular resolution, thereby enabling corresponding HLA antibody assessments. The model prioritised patients according to an established ranking scale based on HLA mismatches. Additionally, the model was completed by establishing an interface platform (Nephron) to create a database for donor–recipient matching. The model can be used as a matching tool for local transplant centres and can also provide remote access for service centralisation. It promoted donor–recipient matching and improved organ allocation accuracy and times. In particular, the model helped resolve the complexities associated with matching HLA systems between different donor–recipient pairs and can be used as a building block to optimise graft survival predictions. Moreover, although the model was mainly developed for kidney allocation, it may be used for any other solid organ allocation system built on similar features at single centres or on a national level.

The key contributions are as follows:The development of a ML-based approach for kidney donor–recipient matching as an alternative to traditional points-based allocation systems.The incorporation of a custom ranking algorithm to identify the most suitable recipient for each donor, potentially improving matching efficiency and outcomes.Nephron is a web-based platform integrating the ML model. Nephron facilitates efficient patient selection and prioritisation, potentially providing wider applications for solid organ allocation beyond the kidneys.The successful implementation of the ML-based approach demonstrates its potential to enhance kidney allocation efficiency and fairness, which are crucial factors given the global shortage of kidney donors.

The remainder of this paper is structured as follows: Section 2 presents the materials and methods employed in the study. Section 3 delves into the three-scenario experiment and discusses its results in detail. Section 4 and Section 5 introduce the automated ranking method and the web-based platform Nephron, respectively. Section 6 provides an overview of the expert evaluation conducted for the Nephron platform. Finally, Section 7 concludes the work presented in this paper.

## 2. Materials and Methods

### 2.1. Dataset Description

The dataset for this study consists of 1000 patients with end-stage renal failure awaiting kidney transplantation collected from King Fahd general hospital, Prince Abdulmajed Dialysis Centre in Jeddah, KSA. The dataset consists of a patient’s age, gender, ABO blood group type, HLA genotype, and HLA antibody specificities. The HLA genotyping was performed using a molecular method at a low/medium resolution designed as (*) followed by a two-digit nomenclature for HLA-A, B, C, DRB1, and DQB1, with a total of 10 HLA types for each candidate, both donors and recipients. Table 1 provides an instance of the data utilised for a single candidate.

The collected data were labelled into five main matching classes, which are outlined below:(1)Perfect match (Figure 1a): In this case, there is a 100% perfect match in all loci of HLA (10/10), a compatible match in terms of ABO blood group (as in Table 2), and the age difference between recipient and donor is less than 20 years (as an example).

(2)Acceptable mismatch (Figure 1b): In this case, there is an HLA mismatch between donor and recipient in the absence of donor-specific HLA antibodies. There is a compatibility in ABO blood group, and the age difference between donor and recipient is less than 20 years.(3)Unacceptable mismatch due to ABO blood group mismatch (Figure 1c): In this case, there is a mismatch between donor and recipient ABO blood groups, which is a contradiction to transplantation. No further matching assessment is required.(4)Unacceptable mismatch due to donor-specific HLA antibodies (Figure 1d): In this category, there are HLA antibodies against donor HLA-A, B, C, DR, or DQ antigens. This assessment is performed for a compatible pair in terms of ABO blood group and age difference.(5)Unacceptable mismatch due to donor–recipient age mismatch: This category was included to resemble the longevity matching criteria between donor and recipient. The acceptable age difference can be adjusted according to the transplant centre’s preference; however, in this model, the acceptable age difference was adjusted to 20 years as an example.

### 2.2. Data Preprocessing

Data preprocessing helps address many challenges inherent in datasets and ultimately leads to the creation of refined and reliable datasets for further analysis. In this study, the following preprocessing steps were applied:(1)Dealing with duplicated and null values:To ensure the reliability and high performance of matching models, the dataset was examined for any duplicated or null values. Duplicate values were resolved by keeping only one instance, and any null values were replaced with the average of observed features related to a corresponding case. Thoroughly selecting suitable replacements guaranteed dataset consistency and suitability for training models.(2)Encoding of categorical values:All features, except for recipient and donor age, consisted of categorical values that necessitated transformation to numerical representations. A one-hot encoder was used to encode blood groups, with a value of one being assigned, and all other categories were assigned zero. The remaining categorical features were converted by parsing their values and extracting numerical values from their categorical representations. An HLA value was typically represented using a combination of letters and digits. In this research, the value was converted to a numerical representation equal to its digit part. For example, A*04 was transformed to the numerical value 4. Likewise, an anti-HLA value was represented by a letter followed by digits and was converted to a numerical value equal to its digit part. For instance, B12 was converted to the numerical value 12.

### 2.3. Development of Matching Models

Developing a range of diverse ML models is essential for ML experimentation. The exploration of different models and techniques allows for assessing their potential to improve performance, ultimately helping to identify the most effective model. Several classification models were developed. Model selection was based on their representation of primary classification modes: linear-based, non-linear-based, decision tree-based, and probabilistic-based. Furthermore, these models are widely recognised as the most prevalent classification models in the medical field. Classification models include the following:(1)Logistic regression (LR) classifier: This linear model assumes a linear relationship between features and the log-odds of a target variable. Its performance was tuned using a grid search technique [38]. Evaluated hyperparameters include the choice of the solver (‘lbfgs’), the number of classes (‘multinomial’), regularisation types (‘l1’ and ‘l2’), and regularisation strength C (0.1, 1, 10). The option to include an intercept term was also explored using ‘True’ and ‘False’ values for the ‘fit_intercept’ parameter.(2)Decision tree (DT) classifier: This tree-like structure makes decisions based on features by recursively partitioning the data based on different attributes and creates a set of rules for classification [39]. It was tuned with the following hyperparameters: maximum depth (10, 20, 30), maximum features (None, ‘sqrt’, ‘log2’), minimum samples per leaf (1, 2, 4), and minimum samples per split (2, 5, 10).(3)Random forest (RF) classifier: This ensemble learning algorithm combines multiple decision trees to improve classification accuracy [40]. The following hyperparameters were tuned using grid search: number of trees in the forest (100, 500, 1000), minimum samples required to split an internal node (2, 5, 10), minimum samples required at each leaf node (1, 2, 4), maximum number of features considered for splitting (‘auto’, ‘sqrt’), maximum depth of the tree (None, 10, 20, 30), and bootstrapping option (True, False).(4)Support vector machine (SVM) classifier: This classifier finds an optimal hyperplane to separate data points into different classes by maximising the distance between the hyperplane and the nearest data points of each class [41]. It can handle both linear and non-linear data using different kernel functions. It was tuned with the following hyperparameters: regularisation parameter C (1, 10, 15) and kernel coefficient gamma (‘scale’, ‘auto’, 0.1, 1).(5)Gradient boosting (GB) classifier: This classifier combines multiple weak prediction models, typically decision trees, to create a strong predictive model. It works by sequentially adding new models that correct the errors made by the previous models [42]. The optimised hyperparameters included the number of estimators (100, 200, 300), learning rate (0.1, 0.2, 0.3), and maximum depth (4, 6, 8).(6)eXtreme gradient boosting (XGBoost) classifier: Similar to GB, XGBoost is a gradient boosting algorithm that iteratively adds weak prediction models and combines their predictions. However, XGBoost leverages parallel processing for significantly faster training on large datasets compared to GB, whose sequential training leads to slower execution times [43]. The following hyperparameters were tuned using grid search: gamma (0.1, 0.2, 0.3), learning rate (0.01, 0.1, 0.2), max depth (4, 6, 8), and number of estimators (100, 200, 300).(7)The CatBoost (CB) classifier: This gradient boosting algorithm excels in handling categorical features. It automatically encodes categorical variables and incorporates advanced techniques for improved performance and efficiency [44]. Grid search was used to tune its hyperparameters for optimal performance: depth (3, 4, 5), iterations (1000, 2000, 3000), and learning rate (0.01, 0.1, 0.2).(8)LightGBM (LGBM) classifier: This gradient boosting framework is known for its high speed and efficiency in handling large datasets [45]. It employs a unique decision tree growth approach that focuses on leafs rather than entire levels. Grid search was used to tune the following hyperparameters: learning rate: (0.01, 0.1, 0.2) and number of estimators (100, 200, 300).(9)The naive Bayes (NB) classifier: This algorithm is based on Bayes’ theorem and operates under the assumption of independence between features [46]. It calculates the probability of a data point belonging to a class based on the probabilities of its individual features. Unlike the previous models discussed, naive Bayes was trained without any hyperparameter tuning.(10)Neural network (NN) classifier: This classifier is inspired by the structure and function of the human brain. It consists of interconnected neurons organised into layers. Each neuron performs a simple computation (weighted sum of its inputs) and passes the result to the next layer using a non-linear activation function [47]. The performance of the neural network was optimised by exploring different hyperparameters: dropout values (0.5, 0.3, 0.35, 0.2), hidden neuron combinations, learning rates (0.001, 0.01, 0.1), and the number of epochs (1000, 1100, 1200, 1500, 2000).

### 2.4. Performance Evaluation

To evaluate the effectiveness of the aforementioned classifiers in predicting successful donor–recipient matches, standard evaluation metrics commonly used in medical classification tasks were employed. These metrics included accuracy, precision, recall, and F1-score [48]. Applying these metrics allows for a reliable determination of which models are most suitable for the task of donor–recipient matching.
(1)Accuracy: This is the percentage of instances successfully predicted out of all instances. It evaluates the overall performance of the model’s predictions [48]:
(1)$$ccuracy=\frac{\left(TP+TN\right)}{\left(TP+TN+FP+FN\right)}$$where
True Positive (TP) is the number of instances in which the model correctly predicted a positive class;False Positive (FP) is the number of instances in which the model incorrectly predicted a positive class (Type I error);True Negative (TN) is the number of instances in which the model correctly predicted a negative class;False Negative (FN) is the number of instances in which the model incorrectly predicted a negative class (Type II error).
(2)Precision: This is the percentage of accurate positive predictions, or true positive cases, out of all positive predictions made by the model. It assesses the accuracy of the model in correctly identifying positive outcomes [48]:
(2)$$Precision=\frac{TP}{\left(TP+FP\right)}$$
(3)Recall: This is the proportion of genuine positive predictions (positive instances that were accurately predicted) to all actual positive instances, also known as the sensitivity or true positive rate. It evaluates how well the model can identify each good occurrence [48]:
(3)$$Recall=\frac{TP}{\left(TP+FN\right)}$$
(4)F1-score: This is the harmonic mean of recall and precision. When both false positives and false negatives are considered, it offers a balance between precision and recall. It measures the compromise between recall and precision [48]:
(4)$$F1-Score=2\left(\frac{Precision\times Recall\;}{Precision+Recall}\right)$$

## 3. Experiments and Results Discussion

Three experiments were designed to comprehensively evaluate the chosen classifiers. The first experiment established a baseline performance level by employing the original dataset with its five distinct matching classes. The second experiment evaluated the adaptability of the selected classifiers to a reduced number of classes achieved by merging the two matching classes. This process aimed to identify classifiers that maintained strong performance irrespective of the complexity of the classification task. Finally, the third experiment leveraged the merged dataset within a hierarchical classification structure. This structure mimicked real-world scenarios by organising classes in a layered fashion. A comparative analysis of these experiments reveals classifiers that exhibited consistent performance across varying levels of classification complexity. For all experiments, the data were split 80:20 for training and testing.

### 3.1. First Experiment (Baseline Original Dataset): Classification with Five Matching Classes

To establish a baseline performance, the first experiment directly evaluated the chosen classifiers on the original dataset. This dataset encompassed five crucial classes reflecting donor–recipient compatibility: perfect match, acceptable mismatch, mismatch due to HLA antibodies, mismatch due to incompatible blood group, and mismatch due to donor–recipient age differences. To identify the optimal configuration for each classifier, a GridSearchCV technique was used. Table 3 showcases the optimal hyperparameter values utilized in this experiment for each classifier.

Figure 2 summarises the performance of all classifiers across the standard evaluation metrics. This visual representation allows for a clear comparison of the classifiers’ effectiveness in predicting successful donor–recipient matches.

The results revealed that gradient boosting methods consistently emerged as the top performers across all evaluation metrics, with accuracy reaching 98% for GB, 97% for LGBM, 96% for CB, and 95% for XGB. This strong performance of the gradient boosting models can be attributed to several key factors, including their ability to handle large datasets effectively, their flexibility in capturing complex non-linear relationships within the data, and the iterative nature of gradient boosting, where the model learns from its errors and progressively enhances its predictive capabilities. The NN classifier also achieved high performance, scoring 95% across the evaluation metrics. The robust performance observed can be attributed to the inherent parallelism of neural network architectures. This parallel processing capability facilitates the efficient handling of large datasets and complex computations, enabling the identification of intricate patterns within the data that are critical for successful donor–recipient matching.

In contrast, the DT and RF classifiers achieved lower accuracy (79% and 80%, respectively) compared to gradient boosting models and NN. Inherent limitations of DT and RF architectures explain the observed performance disparity. These models rely on splitting data based on individual features, restricting their ability to capture complex relationships within the donor–recipient data. Similarly, SVM and LR achieved lower accuracy (79% and 57%, respectively). These models are known for high performance with linearly separable data, a characteristic that this dataset might not possess. The presence of non-linear relationships and complex interactions within the data likely limited the effectiveness of these models.

Finally, the NB classifier exhibited the lowest accuracy (42%). This can be attributed to its core assumption of independence between features, which might not be valid for the dataset employed in this study. The violation of this assumption likely explains the NB’s inferior performance compared to the other classification methods.

### 3.2. Second Experiment (Merged Dataset): Classification with Four Matching Classes

In real-world applications, instances of perfect matches, such as those observed between genetically identical siblings or twins, are infrequent, whereas occurrences of medically permissible mismatches are relatively prevalent. To address the limited availability of “perfect match” data, which could potentially impact model performance, the second experiment merged the perfect match instances into the “acceptable mismatches” class. This resulted in a merged dataset with four classes: the combined acceptable class, mismatch due to HLA antibodies, mismatch due to incompatible blood group, and mismatch due to age differences. Importantly, the merged class maintained a realistic ratio reflecting the true incidence of perfect matches (around 5%) within the broader category of acceptable mismatches. This balancing step ensured the dataset accurately reflects real-world scenarios, leading to a more robust evaluation. Similar to the first experiment, to ensure optimal performance on the merged dataset, the hyperparameters for each classifier were optimised using GridSearchCV, as illustrated in Table 4.

Figure 3 visualises the performance of all classifiers on the merged dataset, as measured by the standard evaluation metrics.

The second experiment (Figure 3) confirms the dominance of gradient boosting methods (GB: 99%, CB: 96%, LGBM: 95%, XGB: 95%), even with a reduced number of classes. Their inherent adaptability facilitated effective management of changes in class structure. This consistent performance across datasets highlights the generalisability of these approaches.

In contrast, neural networks (NNs) exhibited a performance decline from 95% to 90%. Merging classes may have obscured some of the subtle patterns that NNs are adept at identifying, thereby hindering their ability to distinguish between the combined classes.

Interestingly, simpler models (DT, RF, SVM, LR, and NB) demonstrated performance improvements. Reducing the number of classes simplified the data distribution, thereby facilitating the learning of underlying relationships and rendering it easier for these models to discern the fundamental associations. DT accuracy improved from 79% to 89%, RF from 80% to 85%, SVM from 76% to 79%, LR from 57% to 59%, and NB from 42% to 48%. This underscored a key strength of simpler models: their capacity to perform efficiently with less complex data. Figure 4 provides a comprehensive visual comparison of the classifiers’ performance across both the first and second experiments. This visualisation allows for a direct comparison of their accuracies on the original multi-class dataset and the merged dataset, supporting the key insights and findings discussed throughout this section.

### 3.3. Third Experiment: Hierarchical Model

Real-world healthcare settings frequently face a substantial backlog of patients awaiting kidney transplants. This can lead to extended matching times due to the numerous factors that require analysis during the process. To address this challenge, a hierarchical classification model was developed in the third experiment, which aimed to eliminate mismatched candidates using a smaller set of features. This approach has the potential to significantly reduce the overall matching time and complexity, providing a more practical solution for deployment in actual clinical settings.

In this experiment, the classification process was divided into three hierarchical components (Figure 5 provides an overview of the proposed hierarchical model architecture):Component one: blood compatibilityThis component evaluates blood compatibility. If blood groups are incompatible, the result is classified as a mismatch due to incompatible blood group, and the process exits. If blood groups are compatible, the process proceeds to component two.Component two: age compatibilityThis component evaluates donor–recipient age compatibility. If the age difference falls outside the acceptable range (20 years), the data are classified as a “mismatch due to large age gap”, and the process exits. Otherwise, the process proceeds to component three.Component three: HLA compatibilityThis component assesses HLA compatibility. If the recipient possesses HLA antibodies that render the donor incompatible, the data are classified as a “mismatch due to antibodies”, and the process exits. Conversely, if the recipient lacks such antibodies, and the HLA match is suitable, the data are classified as an “acceptable match”, and the process exits.

Leveraging its strong performance in prior experiments, the GB classifier served as the foundation for implementing the three-component hierarchical classification models. To optimise the performance of this hierarchical approach, various numbers of estimators were examined for each individual component: 120 estimators for the first component, 150 estimators for the second component, and 150 estimators for the final component. The first two components achieved an impressive accuracy of 100%, and the final component reached an accuracy of 97%. Overall, the entire hierarchical classification model attained a combined accuracy of 98.10%. Figure 6 illustrates the performance of GB, the top-performing classifier, across all three experiments.

## 4. Automated Ranking for Efficient Donor–Recipient Matching

Ranking potential recipients is crucial, as the matching process may identify multiple compatible candidates for a single donor, necessitating an efficient selection method. To identify the optimal recipient, a thorough assessment of HLA compatibility at the antigen level is essential. A similarity score, calculated based on the number of matching antigens, serves as an indicator of the degree of match. To automate this process and improve efficiency, two methods for similarity ranking were explored: cosine similarity (vector-based) [49] and Jaccard similarity (set-based) [50]. Additionally, a custom similarity ranking algorithm was developed.

### 4.1. Cosine Similarity

Cosine similarity, a method for comparing vectors in a multi-dimensional space, was used to assess the similarity between HLA values of donors and recipients. By representing donors and recipients as vectors, cosine similarity quantifies their similarity by calculating the cosine of the angle between these vectors [49]. However, cosine similarity exhibited limitations in accurately ranking acceptable mismatches. Its effectiveness in this task fell below 50% accuracy.

### 4.2. Jaccard Similarity

Jaccard similarity, a metric for set comparison, calculates the ratio of elements shared by two sets to the total number of elements in both sets. The resulting Jaccard index ranges from 0 to 1, with 0 indicating no similarity and 1 representing identical sets [50]. Similar to cosine similarity, Jaccard similarity also achieved an accuracy below 50% in ranking acceptable mismatches.

### 4.3. Custom Similarity Ranking Algorithm

To address the limitations of cosine and Jaccard similarity in ranking acceptable mismatches, a custom algorithm (refer to Algorithm 1) was developed for HLA similarity assessment between donors and recipients. It compares each donor’s HLA value with the recipient’s corresponding values and calculates the HLA similarity score based on the number of matches. This score quantifies HLA compatibility, allowing for transplant suitability assessment. For example, if a donor’s A1 HLA value matches the recipient’s A1 or A2 value, the ranking score increases. The algorithm’s performance was validated through 40 trials with various donor–recipient combinations. It achieved high accuracy and consistently calculated the HLA similarity score with 100% accuracy.

**Algorithm 1 d67e825:** Calculate Percentage of HLA Similarity.

Require: Donor HLAs, Patient HLAs
Ensure: Percentage of HLA similarity
1:	Initialize *SimilarityPercentage* to 0
2:	***for each*** *hla* in *Donor HLAs* **do**
3:	***if*** *hla **in** Patient_HLAs* then
4:	Increment *SimilarityPercentage* by 1
5:	**end if**
6:	**end for**
7:	Calculate Percentage as $\frac{SimilarityPercentage}{length\;of\;Donor\_HLAs}\times 100$
8:	**Output** Percentage

## 5. Introducing Nephron: A Web-Based Platform for Expert Evaluation of the Intelligent Donor–Recipient Matching System

To ensure the reliability of the proposed intelligent donor–recipient matching system, a dedicated web-based platform, “Nephron”, was developed. This platform was designed specifically for domain experts to conduct meticulous examinations and validations of the system’s functionalities. Subfigures within Figure 7 visually demonstrate Nephron’s capabilities through real-world examples of diverse matching outcomes: perfect matches, acceptable mismatches, mismatches due to HLA antibody incompatibility, ABO blood type mismatches, or age mismatches. It is worth mentioning that the platform can easily adopt additional variables as required by transplant centres for specific recipient sensitisation. These can include HLA sensitisation of the patient, waiting time, transplant urgency, and many others.

## 6. Assessment of Nephron Platform Functionality

Nephron’s functionality underwent a rigorous evaluation through a collaborative review process with kidney transplant specialists at Prince Abdulmajed Dialysis Centre in Jeddah, Saudi Arabia. This collaborative assessment enabled a comprehensive examination of the system’s capabilities, resulting in overwhelmingly positive feedback regarding Nephron’s performance.

This positive feedback serves as a robust validation of Nephron’s effectiveness, fostering a high degree of confidence in its ability to generate reliable results within real-world clinical settings. Furthermore, it underscores Nephron’s potential to revolutionise the kidney transplant donor–recipient matching process, paving the way for its seamless integration into clinical practice.

## 7. Conclusions

In this study, an intelligent ML model for donor–recipient kidney transplantation matching was developed. Using a comprehensive examination and validation of a wide range of ML classifiers that represented primary classification modes, significant success was achieved. The gradient boosting classifier emerged as the top performer, boasting an outstanding accuracy rate of 99% in donor–recipient matching. The dataset also incorporated the most commonly used criteria across kidney allocation systems, including age differences, ABO, and HLA matching. These criteria directly influence transplant outcomes.

Moreover, Nephron, as a web-based platform for donor–recipient matching, enhanced effective recipient registration and matching. Incorporating an intelligent classifier along with a website platform that provides a user-friendly interface facilitating interactions and decision making, Nephron greatly enhances and optimises kidney matching for patients awaiting kidney transplantation. The platform is a useful tool, as it generates a waiting list by registering all patients requiring transplantation from all transplant centres. The proposed approach provides new possibilities to enhance patient outcomes, reduce waiting times, and streamline allocation processes. This development represents a significant contribution to the kidney transplant technology field and may help establish regional or national systems for countries lacking robust kidney allocation systems.

Overall, The ML-based donor–recipient matching model has several advantages over points-based allocation systems. Firstly, it efficiently evaluates matching between different antigens, minimising human errors and ensuring accuracy. The model demonstrates robustness and enables timely selection, reducing the time required for organ allocation. It also provides greater flexibility and adaptability, effectively handling diverse datasets, complex features, and complicated relationships between donors and recipients, potentially improving matching accuracy. Moreover, it continuously learns and refines its predictions by incorporating new data over time. Transplant success, complications, medications, and graft survival data can be included in the model for further analysis and predictions, rendering the model a comprehensive tool for improving transplant outcomes. In contrast, the points-based system relies on predetermined criteria that are often derived from experts or established rules, rendering updates and refinements challenging.

## Figures and Tables

**Figure 1 diagnostics-14-02119-f001:**
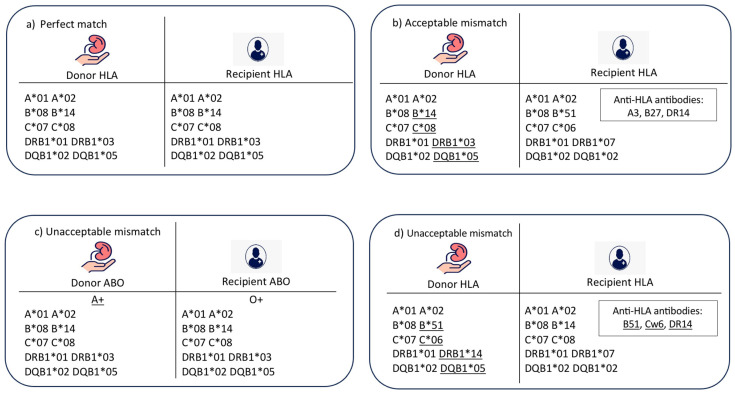
Examples of main matching classes: (**a**) perfect match, (**b**) acceptable mismatch, (**c**) unacceptable mismatch due to ABO blood group, (**d**) unacceptable mismatch due to HLA antibody. * Means the HLA typing was performed using molecular method. The underlines are the mismatch antigens.

**Figure 2 diagnostics-14-02119-f002:**
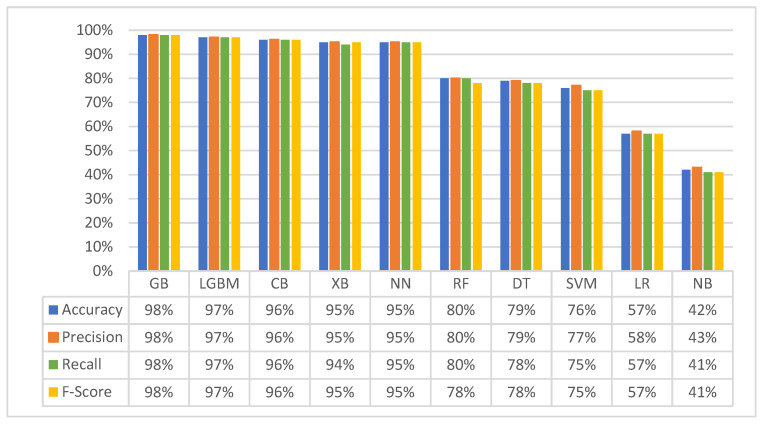
Classifiers’ performance in the first experiment.

**Figure 3 diagnostics-14-02119-f003:**
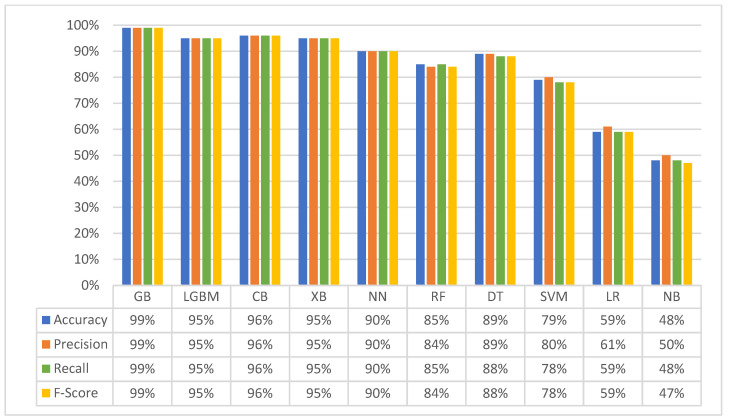
Classifiers’ performance in the second experiment.

**Figure 4 diagnostics-14-02119-f004:**
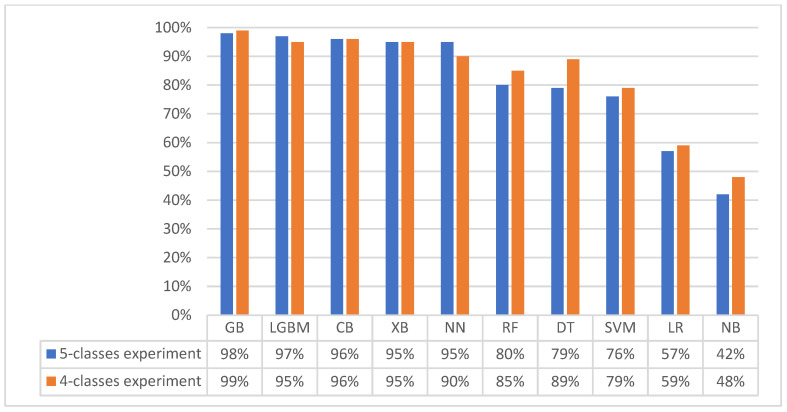
Comparison of classifiers’ performance in the first and second experiments.

**Figure 5 diagnostics-14-02119-f005:**
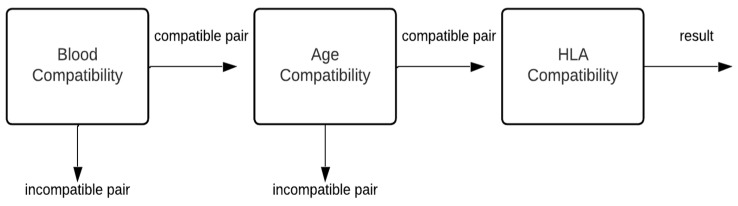
Architecture of hierarchical model.

**Figure 6 diagnostics-14-02119-f006:**
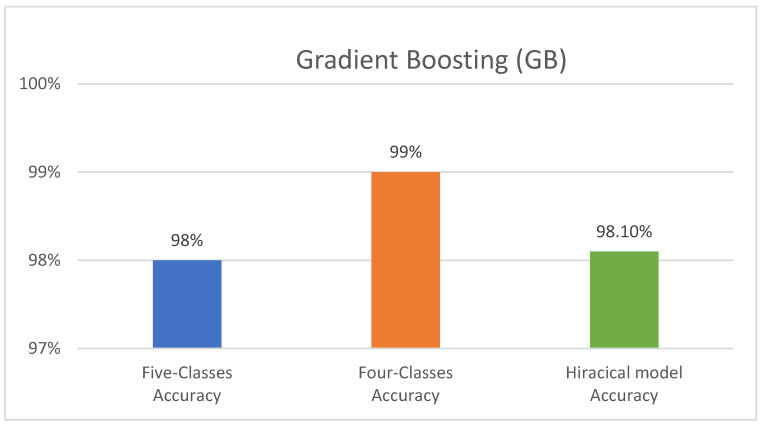
The performance of the Gradient Boosting (GB) model across all three experiments.

**Figure 7 diagnostics-14-02119-f007:**
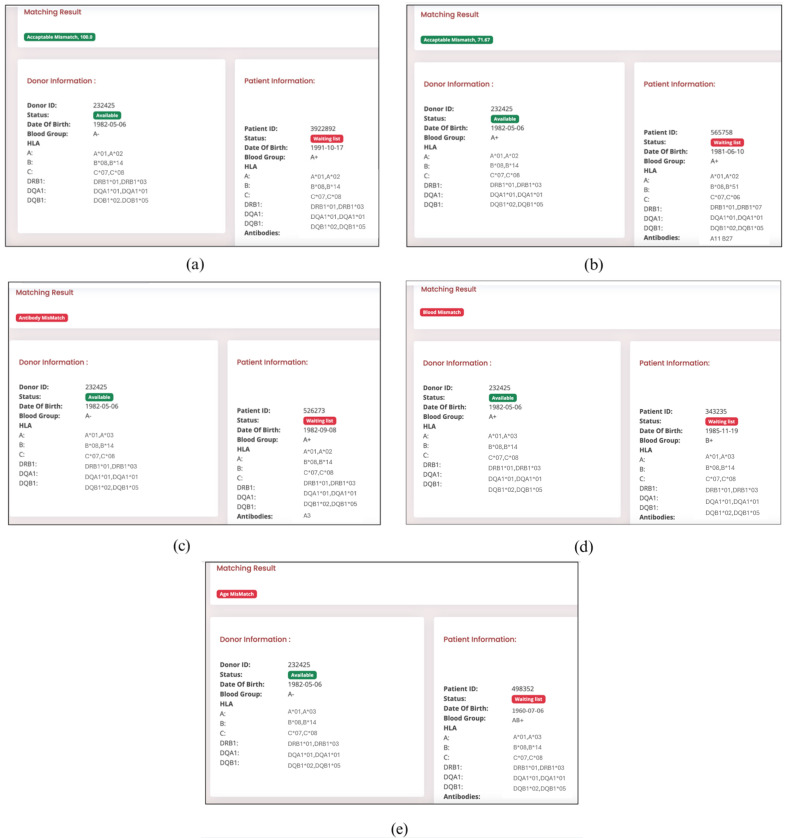
Examples of donor–recipient matching results from the Nephron platform: (**a**) perfect match (acceptable mismatch 100%), (**b**) acceptable mismatch, (**c**) unacceptable mismatch due to HLA antibody, (**d**) unacceptable mismatch due to ABO blood group mismatch, (**e**) unacceptable mismatch due to age mismatch. * Means the HLA typing was performed using molecular method.

**Table 1 diagnostics-14-02119-t001:** Example of data from one candidate.

Characteristic	Value	
HLA-A	A*01	A*02
HLA-B	B*07	B*27
HLA-C	C*03	C*05
HLA-DRB1	DRB1*03	DRB1*11
HLA-DQB1	DQB1*02	DQB1*06
Age	35	
ABO blood group	B+	
HLA antibody specificities (for recipients only)	A3, B38, DQ2, DQ7	

* Means the HLA typing was performed using molecular method.

**Table 2 diagnostics-14-02119-t002:** Matching criteria of ABO blood group between donor (D) and recipient (R).

	D	A	B	O	AB
R					
A		√			√
B			√		√
O		√	√	√	√
AB					√

**Table 3 diagnostics-14-02119-t003:** Optimal hyper-parameters used in the first experiment for each classifier.

Classifiers	Optimal Hyper-Parameters
GB	n_estimators: 200, learning_rate: 0.3, max_depth: 6
LGBM	learning_rate: 0.1, n_estimators: 100
CB	depth: 3, iterations: 2000, learning_rate: 0.01
XB	gamma: 0.2, learning_rate: 0.1, max_depth: 8, n_estimators: 200
NN	hidden layer: (25, 5), learning rate: 0.001, epochs: 15,000, active function: ReLU
RF	n_estimators: 1000, min_samples_split: 10, min_samples_leaf: 1, max_features: ‘sqrt’, max_depth: 10, bootstrap: False
DT	max_depth: 30, max_features: none, min_samples_leaf: 2, min_samples_split: 10
SVM	C: 15, gamma: ‘scale’, kernel: ‘linear’
LR	C: 0.1, penalty: ‘l2’, fit_intercept: true

**Table 4 diagnostics-14-02119-t004:** Optimal hyper-parameters used in the second experiment for each classifier.

Classifiers	Optimal Hyper-Parameters
GB	n_estimators: 200, learning_rate: 0.2, max_depth: 6
LGBM	learning_rate: 0.1, n_estimators: 100
CB	depth: 4, iterations: 3000, learning_rate: 0.01
XB	gamma: 0.2, learning_rate: 0.1, max_depth: 6, n_estimators: 200
NN	hidden layer: (25, 5), learning rate: 0.001, epochs: 15,000, active function: ReLU
RF	n_estimators: 1000, min_samples_split: 10, min_samples_leaf: 1, max_features: ‘sqrt’, max_depth: 30, bootstrap: False
DT	max_depth: 30, max_features: none, min_samples_leaf: 2, min_samples_split: 10
SVM	C: 10, gamma: ‘scale’, Kernel: ‘linear’
LR	C: 0.1, penalty: ‘l2’, fit_intercept: true

## Data Availability

Data is unavailable due to privacy and ethical restrictions.

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
