# Peer review of "Advancing Kidney Transplantation: A Machine Learning Approach to Enhance Donor–Recipient Matching"

_diagnostics, 2024, doi:10.3390/diagnostics14192119_

Round 1

Reviewer 1 Report

Comments and Suggestions for Authors

The manuscript “Advancing Kidney Transplantation: Machine Learning Ap-2 proach to Enhanced Donor-Recipient Matching” describes a valued attempt for development of a machine-learning based approach for donor-recipient matching. I have some suggestions for improvement of the current platform;

1) incorporation of cross-reactive groups (CREG) in acceptable miss-matches seems necessary. On the other hand, eplet matching could provide more suitable candidate donors n the case of mismatches.

2) Recipient’s sensitization and PRA could be incorporated in judgement criteria.

3) It was better to discriminate low-risk and high-risk HLA antibodies in determination of acceptable mismatches as low risk and high risk acceptable.   

4) The figures should be combined and summarized in up to 8 figures.

6) what is the difference and probably superiority of the proposed software over the other comparable platforms such as “OttrOrgan” software?

Reviewer 2 Report

Comments and Suggestions for Authors

The following points need to be addressed in the revision:

1.      The abstract is adequate in length and structure. However, the technical content related to the proposed method is lacking and needs to be improved.

2.      Please thoroughly check your article for typos and grammatical mistakes, like line numbers 55-56, 107, 123.

3.      Add listed contributions to the end of the introduction.

4.      Add the details related to the organization of the article.

5.      Section “C. Development of matching models”, is lacking citations to the works being used in the study. Never forget to pay tribute, to the previous work, in terms of citations.

6.      Why deep learning family of models has not been tried as a possible solution?

7.      Again Section “D. Performance measures” is lacking of citations as fair references. Address this issue.

8.      Use cross-validation techniques to make your results more reliable.

9.      No comments related to the Figure 6 results have been added in the text form (line number 348).

10.  Heading descriptors are not properly organized. Address this issue.

11.  The references related to line number 392 are missing.

Comments on the Quality of English Language

A minor editing would do the job.

Round 2

Reviewer 2 Report

Comments and Suggestions for Authors

The following points are recommended as a minor revision:

1.      The first contribution in the Introduction section carries the word “novel”. I have reservations over the use of this word. Please remove this word, and the rest is fine.
